# Common Functions of Disordered Proteins across Evolutionary Distant Organisms

**DOI:** 10.3390/ijms21062105

**Published:** 2020-03-19

**Authors:** Arndt Wallmann, Christopher Kesten

**Affiliations:** 1Medical Research Council Toxicology Unit, University of Cambridge, Cambridge CB2 1QR, UK; aw889@mrc-tox.cam.ac.uk; 2Department of Biology, ETH Zurich, 8092 Zurich, Switzerland

**Keywords:** intrinsically disordered proteins, protein function, plants, functional comparison

## Abstract

Intrinsically disordered proteins and regions typically lack a well-defined structure and thus fall outside the scope of the classic sequence–structure–function relationship. Hence, classic sequence- or structure-based bioinformatic approaches are often not well suited to identify homology or predict the function of unknown intrinsically disordered proteins. Here, we give selected examples of intrinsic disorder in plant proteins and present how protein function is shared, altered or distinct in evolutionary distant organisms. Furthermore, we explore how examining the specific role of disorder across different phyla can provide a better understanding of the common features that protein disorder contributes to the respective biological mechanism.

## 1. Introduction

Despite the progress made in recent decades, a large proportion of plant protein sequences still lacks useful functional annotation. These proteins represent a promising source for basic research pursuing functional novelty or for translational research seeking new perspectives on biological mechanisms and their disruption in disease. Intriguingly, despite 1.6 billion years of divergent evolution, the majority of genes in *Arabidopsis thaliana* have conserved orthologs in humans, highlighting the similarity of fundamental biological processes between the two organisms [1]. As a result, research on *Arabidopsis* often enhanced our understanding of many molecular mechanisms associated with human diseases like cancer, Parkinson’s disease and Alzheimer’s disease [2,3]. Conversely, homology detection allows for information transfer from the heavily studied mammalian organisms to protein sequences that lack annotation in plants.

Resulting from the recent advances in high-throughput sequencing techniques, the genomes of more than 200 plant species were sequenced, outpacing the more laborious process of experimental protein classification. Hence, only approximately 1% of the protein sequences in the UniProt database have experimentally verified functions [4]. To address this discrepancy, putative or hypothetical proteins are typically classified into protein families that may share evolutionary relationships or molecular function by sequence-based computational analysis. However, this approach works insufficiently for proteins lacking sequence conservation or experimentally verified, functional annotation in orthologous proteins.

## 2. Functional Annotation of Intrinsically Disordered Proteins

Intrinsically disordered proteins (IDPs) and regions (IDRs) lack a well-defined and folded three-dimensional structure in the absence and/or presence of a binding partner. Disorder is a fundamental property of the proteome and can be robustly predicted from primary sequence relying on characteristic patterns of amino acid distribution and overall amino acid content [5,6]. This class of proteins operates largely outside the classic structure–function relationship, with their functionality being defined by a separate set of parameters compared to their structured and globular counterparts [7]. Hence, identification of distantly related IDPs cannot rely on structure, which can help to detect relationships of folded proteins that would remain undetected by conventional sequence-based methods [8]. Methods that use antibodies to metazoan proteins to identify putative plant homologues may result in a higher incidence of artefacts when compared to folded protein targets, due to the smaller size of disordered epitopes and their comparably higher sensitivity to epitope variation [9].

In contrast to folded proteins, IDPs and IDRs show an overall increased rate of evolution, while these rates can strongly vary between different parts of the IDP [10,11]. Moreover, disordered regions appear to be more permissive to mutation, further complicating sequence–function analysis. Overall, genome duplication events seem to influence the distribution of IDRs within genomes, as the amount of identical paralogous IDRs positively correlates with the number of chromosomes [12]. Interestingly, proteins can display different modes of how sequence conservation can relate to disorder as a functional feature [13]. While in some cases only the disorder itself is conserved, but not the sequence facilitating it (flexible disorder), other IDPs retain disorder together with a highly conserved sequence (constrained disorder). Short-linear motifs (SLiMs) represent conserved functional modules that can mediate low-affinity interactions and are often interspersed by regions of flexible disorder [14]. As many alignment algorithms require long stretches of sequence similarity to satisfy statistical significance, the high modality and low complexity of SLiMs can obscure the search for homologous sequences considerably. Due to their limited size (7–12 residues), these motifs can easily develop in unrelated proteins in convergent evolution. Furthermore, post-translational modifications (PTMs) that often regulate IDP function complicate the analysis, as they can drastically alter the biophysical features of a protein and phosphorylation sites display short and weakly conserved motifs that are often difficult to detect via computational sequence analysis [15].

Within protein interaction networks, IDPs often represent hubs carrying multispecific binding sites that adapt to different binding partners [16]. These complexes regularly retain wide conformational flexibility (e.g., fuzzy complexes) even if the binding induces folding in parts of the IDP [17]. Depending on the state of the cell, this flexibility can be rapidly modulated, providing the organism a sensitive framework to respond to changing environments under stress conditions [18]. Indeed, protein disorder is significantly increased in signalling and stress-related processes, providing flexible and rapid adaptation networks for sessile organisms like plants in response to environmental cues [18]. However, despite the numerous IDPs characterized in the animal kingdom and particularly in human diseases, the study of protein intrinsic disorder in plants is still in its infancy.

## 3. Different Frameworks of Protein Similarity

Determining the similarity or difference between entities pre-requires a conceptual system that inevitably creates a hierarchy within sets of features deemed ‘relevant’. The different frameworks of protein similarity, like function, biological context, biophysical properties or evolutionary relationship, will operate on a set of categories that may or may not be compatible with other frameworks. Beyond this, the features within a defined system may not be fixed but draw meaning from reciprocal determination of features (e.g., a protein with a folded domain will likely be classified by that domain rather than by a non-folded, multivalent IDR ‘tail’). Given the ambiguous modes of sequence conservation of disordered regions, it is tempting to assume that the determination of similarity or difference along the lines of function and biophysics will provide a better understanding of the common features that protein disorder contributes to biological mechanisms. However, disregarding evolutionary relationships artificially divorces molecular function from its developmental constraints. In this review, we exemplify the functional kinship of disordered proteins across evolutionary distant organisms and give insights into the essential features that underpin it.

## 4. LEA Proteins as Disordered Spatial Organizers under Cell-Stress Conditions

Although first identified in cotton seeds, Late Embryogenesis Abundant (LEA) proteins are no longer viewed as being restricted to the plant kingdom [19,20]. The members of this family represent one of the most prominent examples of IDP-mediated stress responses in plants [19]. Most notably, LEA proteins accumulate in response to water-deficit conditions and are particularly abundant in anhydrobiotic plants [21]. LEA proteins have been shown to play important roles for the plant responses to salinity, drought, freezing, heat and desiccation [22]. Overexpression studies of selected LEA proteins resulted in increased resistance against diverse abiotic stresses, which underscores their importance as safeguarding proteins [23,24,25,26,27]. Within plants, LEA proteins localize ubiquitously in the cytoplasm and subcellular organelles like the nucleus, mitochondria, endoplasmic reticulum, plasma membrane, chloroplast, and peroxisome [28].

LEA proteins do not display any clear sequence similarity with other proteins of known function and can be classified into several broadly defined families or classes based on conserved protein domains [29], the occurrence of defined sequence motifs [28] or their physico-chemical features [30]. Overall, their amino acid composition is dominated by hydrophilic residues, charged amino acids, small amino acids like glycine or serine and a high proportion of disorder-promoting amino acids, although there is some significant variation within the LEA protein family regarding overall hydropathy and charge (Figure 1 A). Indeed, experimental and bioinformatic evidence suggests that a key feature of the LEA protein family is their total or partial lack of stable structure, resulting in a flexible conformation [31]. Some bacterial and plant LEA proteins also carry folded domains like the Water stress and Hypersensitive response (WHy) domain, which is involved in the desiccation response and may originate from an ancestral domain in Archaea [32,33]. As seen in other disorder-containing protein families, LEA proteins display a diverse multifunctionality, for example as protein and membrane chaperones, as DNA-binding proteins, or sequesters of metal ions and radicals [20]. As many interactions and functions have only been characterized using in vitro approaches, the full scope of this family’s functional repertoire remains obscure.

Previous studies showed LEA proteins to be sufficient desiccation protectants both in vivo and in vitro [34,35]. Indeed, LEA proteins exhibit increased expression during the desiccation-tolerant stages of plant seed development and in dividing cells of root tips [36]. Leaves of Arabidopsis are not desiccation tolerant, but experimental evidence suggests that the LEA proteins COR15A and COR15B assert a cryoprotective function by mediating freezing-induced crowding effects [37]. In vitro experiments showed that hydrophilins like the LEA protein family together with trehalose, or other disaccharides, can vitrify, which directly protects cells from protein aggregation and membrane disintegration [38,39,40]. The formation of these intracellular, glass-like matrices is hypothesized to prevent non-specific protein–protein interactions that otherwise lead to denaturation and aggregation, and thus protect seeds and roots during drought. The general mechanism was proposed to be mediated by the protein’s ability to acquire secondary structure during drying, when intra-protein hydrogen bonds become energetically more favourable [41,42,43,44]. However, the structural aspects of the mechanism have yet to be validated in vivo. In recent years, in-cell solid and solution state, Nuclear Magnetic Resonance (NMR) methodologies have advanced to the point that experimental approaches to study structural transitions under physiological conditions are now conceivable [45,46].

Besides plant organisms, LEA proteins have also been found in other desiccation-tolerant organisms, such as bacteria, fungi, and some invertebrates, suggesting a common mechanism across distinct life forms [47]. However, the full scope of LEA involvement in desiccation tolerance across species is hard to estimate since relatedness identified by sequence-based methods can be compromised given the low sequence complexity of IDPs. Moreover, many organisms carry multiple types of LEA proteins that display diverse subcellular localisation and divergent functions within different species and cellular contexts. Typically, a desiccation-tolerant state is triggered by pre-exposure to slow drying that induces changes in the expression profile of the cell (Figure 1C). Thus, the functional annotation of many LEA and LEA-like proteins was often based on gene expression analyses under stress conditions. These include the two divergent LEA-like proteins (*Ar*LEA1A, *Ar*LEA1B) in the freshwater rotifer *Adineta ricciae* [48], or the desiccation-induced expression of *Aav*LEA1 in the nematode *Aphelenchus avenae* [49]. Another example are the desiccation-tolerant larval stages of the brine shrimp *Artemia franciscana,* in which mRNA levels of *Afr*LEA1 and *Afr*LEA2 transcripts are increased 7- and 14-fold when compared to the stages that do not share the capacity for anhydrobiosis [50]. Although LEA proteins are not represented in mammalian genomes, *Afr*LEA2 transfected into human HepG2 cells enhanced desiccation tolerance in the presence of intracellular trehalose, and resulted in increased membrane integrity after rehydration [51]. Similarly, eight out of fifteen *Arabidopsis* LEA proteins increased tolerance to desiccation when heterologously expressed in *Saccharomyces cerevisiae* [34]. Furthermore, the disordered yeast hydrophilin HSP12 alleviated the damage caused by severe water loss, indicating a synergistic and independent activity together with trehalose [52]. Interestingly, although HSP12 shares the general biophysical features (size, charge, disorder) of the eleven other hydrophilins in yeast, it appears to be unique in mediating desiccation tolerance amongst the hydrophilin family.

Similar to rotifers, tardigrades display desiccation tolerance despite naturally lacking trehalose [53]. Only recently, tardigrade-specific IDPs were shown to vitrify in vitro and in vivo, when heterologously expressed in yeast [54]. Moreover, RNA interference experiments reduced survival to desiccation independently of any sugar mediator. The authors proposed that tardigrade-specific IDPs mediate desiccation tolerance by protecting proteins against denaturation, trapping them in an amorphous matrix [54]. Similar to the examples above, these proteins were identified using differential gene expression analyses of hydrated and slowly drying tardigrades. Interestingly, these mechanistic similarities between tardigrade-specific IDPs and LEA proteins seem to have developed independently in convergent evolution, highlighting the importance of IDPs in organizing cellular matter in response to stress across phyla.

## 5. Protein Disorder as a Driving Force for Liquid–liquid Phase Separation

A classic example of spatio-temporal separation of biochemical processes in eukaryotic cells is the formation of canonical membrane-enclosed organelles. However, eukaryotes also contain numerous membrane-less compartments (MCs), such as the nucleolus or P-bodies [55]. MCs have liquid-like properties and undergo dynamic liquid–liquid phase separation (LLPS). This behaviour enables them to rapidly form on demand, fuse, shear, exchange their content, or disassemble and thus concentrate proteins and biochemical reactions at distinct locations when needed [56,57].

To achieve LLPS and formation of MCs, their contents have to reach a critical concentration to enable a de-mixing effect and form non-miscible phases [58]. Under these conditions, intermolecular interactions between IDPs stabilize the condensed phase overcoming intramolecular and solvent interactions [59,60]. The level of anisotropy remarkably increases in phase separated droplets, indicating an expanded state of IDPs in comparison to a more compacted state in solution [59,60] (Figure 1D). The resulting MCs are highly concentrated in proteinous components and can also rapidly dissociate once the concentration falls below the critical point or protein interactions are altered (e.g., by PTMs) [61,62,63,64,65]. Indeed, the lack of defined structure renders IDPs more accessible to regulation through PTMs that can change the biophysical properties of IDPs and consequently modify protein–protein interaction [64].

Almost all MCs contain a large proportion of IDPs [66,67] and their molecular features were suggested to be essential for LLPS and thus MC formation [67,68]. The lack of defined secondary or tertiary structure and thus high flexibility might provide the needs for the dynamic behaviour of MCs [67,69]. Furthermore, the capacity of IDPs to exhibit promiscuous protein–protein interactions [70] might permit the spontaneous and reversible formation of sufficient, local protein concentrations to initiate LLPS. These multivalent protein interactions are proposed to be mediated by repetitive sequence elements that result in overall IDP sequence simplicity [57,64,71]. A prominent example of an MC-forming IDP is the Essential Pyrenoid Component 1 (EPYC1). It is an indispensable part of the Pyrenoid, a MC in the chloroplast of many algae that concentrates components of the carbon fixation machinery to increase its efficiency [72]. EPYC1 contains four sequentially simplistic regions that have been proposed to form weak multivalent interactions with Rubisco and thus might be central to the liquid-like properties of the pyrenoid [72]. Other examples from the mammalian and fungal kingdoms are DEAD-box helicase 4 (DDx4) proteins that harbour clusters of FG and GF repeats [73] or the polyQ tract of Whi3 [74], respectively.

MCs of plants and other eukaryotes contain overlapping sets of proteins, indicating a common function in eukaryotic metabolism. For example, stress granules, small cytoplasmic MCs that form upon a variety of stresses contain RNA, translation initiation factors, RNA-binding, and other proteins [75,76]. Curiously, also components of the cell cycle regulation machinery (see paragraph below), such as cyclin-dependent kinases localize to stress granules in both plant and human cells [76,77]. However, plants and green algae also contain specific MCs. The chloroplasts of green algae contain the above mentioned pyrenoid and plant photobodies contain light receptors and signalling proteins [78]. Cryptochromes (see paragraph below) are components of photobodies and undergo rapid LLPS upon blue light perception and are thus a valuable optogenetic tool in mammalian cells [79].

Members of the SR protein family were recently shown to adopt similar functions in plant and mammalian MCs. SR proteins are intrinsically disordered, exhibit RNA-binding capacity, are involved in alternative splicing, and contain long repeats of serine and arginine residues [80,81]. The plant-specific SR45 selectively accumulates in nuclear body MCs, in a temperature- and phosphorylation-dependent manner [82]. It was suggested that plant SR proteins might regulate splicing activity in response to stress by undergoing LLPS and thereby concentrating the splicing machinery into MCs [56]. Similarly, the mammalian SR protein SRSF9 was recently shown to regulate nuclear stress body formation upon heat exposure, depending on the phosphorylation state of the protein [83,84]. SR45 and SRSF9 can adopt similar functions in plants and mammals, despite significant differences in protein size (414 vs 221 residues, respectively) and no sequence conservation except in the conserved RNA recognition motifs. Overall, information on plant IDPs that undergo LLPS is still scarce but investigating the composition of plant-derived MCs may represent a promising path towards a better understanding of the fundamental and species-specific features of membrane-less compartmentalization in cell biology. Moreover, features like prion-like domains are important predictors of LLPS in RNA-binding proteins [85]. In *Arabidopsis*, nearly 500 proteins were predicted to carry such domains [86]. Indeed, the plant-specific Flowering Locus CA and FLL2 proteins have recently been shown to be in vivo regulators of LLPS within the autonomous flowering pathway of *Arabidopsis* [87].

## 6. The Role of Disordered Proteins in Microtubule Organisation

Beyond their roles in the structural integrity and division of the cell, plant microtubules also adopt a sensory function for the perception of abiotic stress conditions [88]. Recently, we showed that the intrinsically disordered region of the Cellulose synthase Companion (CC) proteins is critically involved in the salt-stress response of *Arabidopsis* [89]. Representing one of the primary responses to salt stress, the plant’s cortical microtubule network is re-structured and rendered stress tolerant under saline conditions [90]. Plant cortical microtubules steer the movement of the cellulose synthase complex and thus are essential for the organism’s morphology and growth [91]. The CC protein family is an essential player in the microtubule re-assembly process during salt exposure [92]. Interestingly, the mechanism by which the cytosolic N-terminus of CC1 regulates and interacts with microtubules appear to be remarkably similar to that of the human Tau microtubule-associated protein, which is widely known for its potential role in multiple neurodegenerative diseases [89,93]. Both proteins are intrinsically disordered and contain multiple short hydrophobic microtubule-binding motifs that can bind tubulin and microtubules transiently and independently [94] (Figure 2).

Thus, both proteins can promote polymerization and bundling of microtubules, while also being able to diffuse along the microtubule lattice [95,96]. While Tau controls microtubule dynamics and organisation in neurons [97], the CC1-mediated microtubule bundling may underpin microtubule array stabilization of the plant cell during salt stress. Notably, microtubule-associated proteins display an increased content of disorder in eukaryotes [98]. The observation that key biophysical and functional properties are shared across distantly related kingdoms may spark new perspectives on the evolution of microtubule-associated IDPs and their function in stress and disease-related processes.

## 7. The Multivalent Role of Protein Disorder in Cryptochrome Signalling

Cryptochromes (CRYs) are blue-light receptors that regulate varying functions such as cell growth and circadian rhythm in a range of organisms like plants, insects and bacteria. The conserved and folded N-terminal domain of all CRYs resembles photolyases (Photolyase homology region (PHR); [99]). As such, CRYs are flavoproteins but lost their photolyase activity and hence are not involved in DNA damage repair [99]. Nearly all C-terminal extensions of CRYs are predicted to be disordered and vary greatly in length and sequence between species [100,101]. Nevertheless, the function of cryptochromes is critically dependent on these C-terminal IDRs [102,103,104,105,106]. *Arabidopsis* harbours two cryptochromes (CRY1 and CRY2), whose C-terminal IDRs differ in sequence, but are functionally equivalent as they can be interchanged [107,108]. Proteolytic analysis of human and *Arabidopsis* CRYs revealed that the C-terminal IDRs show increased susceptibility to proteolytic digestion after illumination, which suggests a conformational change that exposes the IDR [101]. This observation is in line with the crystal structure of full-length *Drosophila* cryptochrome, which revealed that the flexible IDR resides in a grove of the PHR domain in the non-excited state [109,110]. Transgenic *Arabidopsis* plants overexpressing the CRY C-terminal extension phenotypically show a constant light response, indicating that it is sufficient to activate the otherwise light-induced signalling pathway [102]. Indeed, it directly interacts with several downstream regulatory proteins such as Constitutive Photomorphogenic 1 (COP1) or Suppressor of Phytochrome A1 (SPA1) [111,112,113,114] and is heavily phosphorylated upon light perception [115,116,117]. Figure 3A summarizes the general action model of plant CRYs, which includes the following steps: 1. In the dark, PHR and the C-terminal IDR interact and form a closed conformation; 2. Light induces a dimerization of CRYs and phosphorylation of the IDR leads to an exposed IDR conformation; 3. The IDR binds to regulatory proteins that subsequently modulate developmental processes [101,113,114,116,117,118].

The circadian clock regulation in *Drosophila* requires the tightly regulated binding of multiple proteins to CRYs, e.g., Timeless (TIM) and Jetlag (JET) [119,120]. In response to light, the C-terminal IDR is released from the PHR, which then allows binding of both TIM and JET to the PHR [109,119,120]. In contrast to plant CRYs, the IDRs of *Drosophila* CRYs thus do not facilitate binding to regulatory proteins but inhibit it (Figure 3A). Consistent with this conclusion, the *Drosophila* PHR domain, deficient of its C-terminal IDR, induced a constitutive light response, in contrast to plant CRYs that transmit their response directly via their disordered C-terminus [102,121].

In contrast to both plant or insect CRYs, their mammalian orthologs lack any known direct photosensory function [122,123]. However, mutant mice deficient in the photoreceptors opsin or melanopsin show impairment in circadian clock regulation, indicating that mammalian CRYs are dependent on light perception of other photoreceptors [124,125]. Indeed, they are part of a transcription/translation feedback loop to establish the circadian rhythm. In a complex with Period (PER), they repress the activity of the circadian transcription activator complex circadian locomotor output cycles kaput (CLOCK)/brain and muscle Arnt-like protein 1 (BMAL1) and thus repress their own transcription [126,127,128]. The C-terminal IDR extension of mammalian CRYs regulates the import of the protein into the nucleus as well as the interaction of the PHR with the CLOCK/BMAL1 complex (Figure 3B) [99,105,106]. Interestingly and reminiscent of plant CRYs, phosphorylation of the C-terminal IDR plays an important role to modulate mammalian CRY proteins. Phosphorylation of the mammalian CRY1 IDR stabilizes the protein [129], while phosphorylation of the CRY2 IDR destabilizes the protein and leads to its degradation [130,131]. Curiously, the C-terminal IDR of the mammalian-like CRY ortholog from the green algae *Chlamydomonas reinhardtii* was recently suggested to bind to its PHR upon light perception [132]. Thus, a similar mode of action might be possible for mammalian CRYs, potentially through a light dependent signalling pathway in conjunction with other photoreceptors.

In summary, despite having a conserved PHR domain, the members of the CRY protein family have adopted varying, species-dependent functions, which are mediated by their highly variable C-terminal IDR extensions and may have developed independently. The CRY proteins are thus an excellent example of how protein disorder, despite the presence of other, folded regions with a conserved mechanism, can represent the primary determining factor of molecular function between kingdoms of life.

## 8. Disordered Proteins Represent Key Regulators in Cell Cycle Progression

The cell cycle is one of the most intensively studied processes in biology, especially due to its misregulation in many human diseases. Unlike animal development, plants largely develop post-embryonically and, thus, organ formation, like flowers, leaves, stems, or roots, continuously develops throughout the plant life cycle. Plant cell division is located in meristems, containing pluripotent stem cells whose progeny is subsequently developing into specialized cells. Despite these striking differences in developmental organisation, all eukaryotic cells essentially undergo the same cell cycle that is defined by characteristic phases. Cyclin-dependent kinases (CDKs) play an essential role in the progression of the cell cycle and are conserved in all eukaryotes. A multitude of CDK–cyclin complexes control the transition from the post-mitotic gap phase (G_1_) to the synthetic (S) phase and second gap phase (G_2_) to mitosis (M) phases by phosphorylating downstream target proteins [133]. Cyclins hereby act as mediators between the CDKs and multiple substrates. Because of the essential role in the continuation of the cell cycle, CDK–cyclin complexes are heavily regulated by several mechanisms, like phosphorylation and proteolysis initiated by ubiquitination, all of which were reviewed elsewhere [133,134,135].

Cyclin-dependent kinase inhibitors (CKIs) bind CDKs and inhibit their activity to regulate the progression of the cell cycle. Consequently, CKI misregulation is associated with a multitude of diseases [136,137,138,139]. CKIs are IDPs that only share a conserved inhibitory domain (CID), which acquires a folded state when involved in cooperative binding to both CDKs and cyclins [140,141,142,143]. The budding yeast CKI SIC1 and its mammalian counterparts of the p27^kip1^ family share very low sequence homology [141]. However, prediction tools indicated structural similarity of the CIDs. Intriguingly, the positioning of the CID within the overall topology of the CKIs varies between different species. While the domain is located at the C-terminus of SIC1, its position is N-terminal in p27^kip1^ (Figure 4A) [140,144]. Despite these positional differences, heterologously expressed SIC1 can bind and inhibit the activity of the Cdk2–cyclin A complex, the mammalian binding partner of p27^kip1^, in vitro and overexpression of mammalian p27^kip1^ in a SIC1 deficient yeast strains can rescue its cell cycle related phenotype [141]. The first plant CKI was identified by a yeast two-hybrid assay employing *A. thaliana* CDKA;1 as a bait protein and subsequent sequence analysis, in which the CID was found to be distantly homologous to that of mammalian p27^kip1^ [145]. This Kip-related protein (KRP) family subsequently expanded to seven members in *Arabidopsis* [146,147] and fulfils an analogous function in cell cycle control as their yeast and mammalian counterparts [147,148,149,150]. Curiously, like the yeast protein, they carry the conserved inhibitory domain at their C-terminus (Figure 4A) [145,147,149] and heterologously expressed *Zea mays* KRPs furthermore decrease overall cell size when expressed in fission yeast [151], hinting at a delay in cell cycle progression. In contrast to the functional conservation between yeast and plant CKIs, classic sequence alignment of the CIDs show no homology better than expected by chance, while KRPs and mammalian p27^kip1^ are at least distantly related [152]. However, the CIDs of all three CKIs acquire or are predicted to acquire a common fold of two α-helices interspaced by a flexible linker (Figure 4B).

The IDR domains of the CKIs presented here, especially p27^kip1^, have been associated with many functions, e.g., transcriptional regulation, regulation of the cytoskeleton or tumour development, but these were reviewed elsewhere [153,154,155]. One of the most investigated functions is the regulation of their own CID domain. To release CKIs from their respective CDK–cyclin complexes and thus activate them, phosphorylation of the IDR domains at multiple positions is required. The process is best described in budding yeast. SIC1 is tightly bound to Clb (B-type cyclins)-CDK complexes at the beginning of the cell cycle and therefore inhibits them [156,157]. To initiate deactivation of SIC1 through ubiquitination and subsequent degradation through the 26S proteasome [158,159,160], at least six out of multiple phosphorylation sites in its IDR domain (Figure 4A) have to be phosphorylated in a specific cascade [161,162,163,164]. Yeast cells enter the (S) phase of their cell cycle afterwards. Furthermore, phosphorylation of other, differential positions in SIC1 seems to play a role in cell cycle progression [165,166].

The mammalian analogue p27^kip1^ is also heavily regulated by phosphorylation. An overview of important p27^kip1^ phosphorylation sites can be found in Figure 4A. While phosphorylation of residues S10 and T198 is mainly involved in protein stability in interphase cells [167,168,169], together with T157 they also contribute to the subcellular localization of the protein [170]. When progressing through the cell cycle Y88 gets phosphorylated, which partially releases p27^kip1^ from the CDK2/cyclin A complex [171]. Further phosphorylation of Y74 leads to CDK2 activation and phosphorylation of T187 in p27^kip1^ in an intra-complex manner [172,173]. This leads to deactivation of p27^kip1^ through ubiquitination and subsequent degradation through the 26S proteasome [174] and ultimately entry into (S) phase [171].

The regulation of plant KRPs is not well understood to date. However, KRPs are phosphorylated before deactivation, although the key sites remain unknown [175]. Phosphorylation in all three protein classes is thus an important regulatory factor. Indeed, SIC1 acquires different transient folding states when being phosphorylated or dephosphorylated, influencing the electrostatic field experienced by the SIC1 binding partner [176]. Furthermore, a region in p27^kip1^, in between the important phosphorylation sites Y88 and T187, shows a charge pattern that is important for the selective phosphorylation of T187 in the cascade of events leading to deactivation of p27^kip1^ and (S) phase entry [177]. Remarkably, SIC1 exhibits a comparable charge pattern in between annotated, important phosphorylation sites. Furthermore, predicted phosphorylation sites in KRP1, identified with the NetPhos 3.1 server [178], frame a region with such a charge pattern. The region thus seems to be functionally conserved through biophysical properties, despite no classic sequence conservation (Figure 4C).

## 9. The Role of Protein Disorder in Transcriptional Regulation

While the general principles of protein–DNA recognition are well conserved among eukaryotes, transcriptional control that relies on protein–protein interactions is more species-specific [179]. This situation is further complicated by the fact that these regulatory domains often employ disordered SLiMs that are difficult to robustly predict de novo from sequence and can develop functionality through both rapid convergent and divergent evolution [180]. The overall disorder content of TFs may be linked to the capacity of to establish more complex gene regulatory networks in multicellular organisms [181]. In *Arabidopsis thaliana,* a large proportion of transcription factors (TFs) contains extended regions of disorder (82%–94%, [182], disorder in eukaryotic TFs is reviewed in [183]). In particular, their transactivation domains (TADs), scaffolding domains responsible for recruiting transcriptional co-regulators that are critical for transcription initiation, display a high degree of disorder (73%–95%, [182]). The “Nine amino acid Transactivation Domain” (9aaTAD) family is a prominent example of an important generic TAD in eukaryotes. Its motif is defined by a tandem of hydrophobic clusters, hydrophilic residues with proportional positive/negative charge and a 3 aa hydrophobic region towards its N-terminus [184]. Despite showing wide variability across species, the 9aaTAD represents a universal module that mediates binding to the transcriptional machinery. For instance, plant (e.g., MYB63), yeast (e.g., Oaf1p) and animal (e.g., SREBP) TFs were found to harbour 9aaTADs that facilitate interaction with the Med15 KIX domain of the Med Mediator Tail module [185,186,187].

Understanding the mechanistic details of TAD recruitment promises the opportunity of protein design for crop optimization in food and feed production due to their prominent involvement in plant stress responses. TAD engineering by design principles taken from other organisms was applied early on in plant research. Many studies used the viral protein 16 (VP16) acidic activation domain from the herpes simplex virus to change transcript levels of specific target genes [188,189]. For instance, a zinc finger DNA-binding protein-VP16 fusion construct targeting the b-KETOACYL- ACP-SYNTHASE II allowed for the modification of the oil content in rapeseed leaves and seeds [190]. Much like the TAD of human tumour suppressor p53, the VP16 TAD contains two disordered activation subdomains, each with transcription activation potential that may form amphipathic α-helices upon complex formation [191,192]. Recently, Krois and co-workers gave remarkable structural insights into the binding specificity of p53 by showing how the p53 TADs directly compete with non-specific DNA sequences for binding to the DNA-binding core domain [193]. As neither VP16 nor p53 are present in the genome of vascular plants, the VP16 design principles were used to screen for TADs of plant regulatory regions [194]. Interestingly, some of the identified domains significantly improved transcriptional activation and exhibited higher efficacy in planta when compared to VP16. Approaching the design challenge in a high-throughput manner, Ravarani and co-workers developed an IDR-Screen framework for TADs [195]. Using a yeast transcription factor assay with Heat shock factor protein 1 as a bait protein, the authors screened a random sequence library and variants of known TADs to derive sequence patterns that underlie TAD function. The surprisingly large amount of functional TAD sequences was enriched in negatively charged amino acids and aromatic hydrophobic residues. However, also highly degenerate and redundant sequences were sufficiently functional within the assay, which may indicate broad compatibility with co-factor interaction and/or non-specific binding to the components of the transcriptional machinery. Moreover, the high sequence degeneracy of the TAD sequences hints for a binding mechanism that is primarily mediated by multiple and fuzzy interactions rather than strong specific binding.

With representatives in over 100 land plant species and over 100 genes in *Arabidopsis* alone, the NAM/ATAF1/CUC2 (NAC) family is one of the largest plant-specific transcription factor families and of vital importance in the stress response and cell wall synthesis of the plant organism [196,197]. These proteins contain a conserved and structured N-terminal DNA-binding domain, while a highly variable C-terminal domain is predicted to be largely disordered [198,199]. NAC proteins interact with a number of different proteins but it is yet unclear which are mediated by the disordered C-terminal domain [200,201]. Upon complexation with the stress regulator and hub-protein Radical-induced Cell Death 1, the C-terminal domains of ANAC046 and ANAC013 do not adopt an induced structure, which is consistent with an inherent conformational flexibility and fuzziness of the interaction [202]. The Suppressor of Gamma Response 1 (SOG1) is a plant-specific NAC transcription factor that regulates the DNA damage response [203]. Similar to p53, SOG1 regulates a variety of genes involved in cell cycle arrest, apoptosis, DNA damage response and repair. Due to these similarities and since the plant genome lacks a p53 ortholog, SOG1 has been put forward as a functional analogue of p53 [204]. Indeed, in an analysis of the *Arabidopsis* DNA damage response transcriptional network, SOG1 represented a major activator targeting other TFs, repair factors, and cell cycle regulators and thus coordinates the induction of DNA damage repair [205]. Interestingly, many of the identified genes targeted by SOG1 have human orthologs that are p53 targets and thus share a similar role within their respective regulatory networks. One example is the SOG1 target KRP6, a CKI resembling p21 and p27 (see above), which conversely represents a major target of p53 activity and mediates the down-regulation of cell cycle genes [206]. Despite functional similarities, the domain architecture and sequence are not conserved between p53 and SOG1, suggesting that the kinship of the two proteins is not rooted in common ancestry, but may have developed independently in response to unique demands imposed by their respective species’ DNA damage repair networks. Similar to the NAC family members described above, the SOG1 C-terminal domain is predicted to be disordered and appears to be strongly post-transcriptionally regulated [207]. Hence, it is conceivable that a disordered C-terminus may enable SOG1 to display a broad functional and structural repertoire similar to p53, which interacts with a large number of protein partners. The much lower number of confirmed SOG1 interaction partners in comparison to p53 may stem from the less extensive characterization of the plant DNA damage repair mechanism.

## 10. Methodological Advances and Outlook

From the shape of the cell, the ultrastructure of the cytoskeletal network, down to protein structure at atomic resolution, structural observations have traditionally been the predominant framework of comparisons between the molecular life of plant and metazoan organisms. As the conservation of proteins is more pronounced at the structural than the sequential level, IDPs and IDRs thus represent a challenging target for functional comparison [208]. Computational approaches of high-throughput protein functional annotation are highly desirable to guide more in-depth and laborious in vitro and in vivo analysis. Besides structural considerations, current computational protein function predictions rely either on sequence or information-based methods [209]. In IDPs, the sequence–function relationship is often independent of structural restraints and thus requires novel methods for analysis. Applying an average distance map technique, Shimomura and co-workers were able to identify disordered regions that show a tendency to adopt an ordered structure in their bound state [210]. Zarin and co-workers could show that biophysical features like net charge or hydrophobicity of amino acids rather than sequence seem to be the determining factor in the evolution of IDPs [211]. The authors could demonstrate that intrinsically disordered regions can be readily exchanged based on their physical properties without the need for classic sequence conservation. Interestingly, the pattern of protein disorder itself was put forward as an alternative approach to trace distant relatives in classes of proteins with high levels of intrinsic disorder [212]. These patterns may have developed in response to species-specific requirements and biological context and will thus likely aid in establishing the key determinants of the molecular mechanism. Disregarding the premises of the sequence–structure–function relationship altogether, information-based methods are well suited to identify disordered proteins of analogous function in distantly related phyla. Recent advances made in high-throughput transcriptomics at single-cell resolution and the analysis of elaborate co-expression networks will allow researchers to gain insights into complex regulatory relationships and identify novel players in the targeted biological processes [213,214]. Multiple methods have been developed that exploit protein–protein interaction networks in order to identify protein function based on the topological features of the target proteins interaction network independent of structure or sequence homology [215,216]. Hence, the ever-increasing amount of protein–protein interaction and co-expression data may help to illuminate common biological mechanisms of proteins between distantly related phyla by comparing their position within the respective networks. However, despite recent improvements in the analysis, identifying hub proteins is critically dependent on the quality and origin of the underlying data [217].

The arsenal of biophysical methodologies to characterize structural features of IDPs has significantly expanded in recent years and thus enables a more refined functional comparison between proteins from different phyla. The integration of experimental data from NMR, small-angle X-ray scattering and molecular dynamics simulations can yield a detailed structural description of IDP conformational ensembles [218,219]. Although the capacity to resolve protein disorder with cryogenic electron microscopy (cryo-EM) remains limited, the method can make significant contributions to elucidate IDP binding to higher-order molecular complexes [220]. Magic-angle spinning (MAS) NMR carries the potential to study both rigid and flexible protein regions and can also be applied to study structural properties in living cells although plant cells pose a special challenge for protein delivery due to their cell wall [46,221,222]. Beyond the improvements in high-throughput phosphoproteomics by mass spectrometry, time-resolved solution-state NMR methods have been developed to probe phosphorylation patterns in cell extracts, intact cells, or under defined physiological conditions [223,224]. Furthermore, elaborate single-molecule and microfluidic techniques can describe the collective properties of IDPs in LLPS or during oligomerization [225,226]. Characterization of single molecules and their dynamic behaviour in vivo is still challenging. Live cell confocal imaging based on single-molecule Förster resonance energy transfer (FRET) is an established tool to study biophysical features of IDPs in vitro [227,228], but was recently also proven to be a versatile tool to study biophysical features of IDPs in vivo, e.g., dimensions, submicrosecond chain dynamics or conformational changes upon interaction with binding partners [229,230,231,232]. However, plant proteins are usually imaged at the organismal level and imaging is thus based on expression of fluorescent fusion proteins, rather than on microinjected proteins that are chemically linked with fluorescent dyes. Due to weaker photophysical features of these fusion proteins, their application for extended single molecule FRET is limited [227]. Additionally, fusion proteins are bulky molecules in comparison to fluorescent dyes and might influence the accuracy of measurements. Introduction of non-canonical amino acids and subsequent click chemistry based labelling [233,234] of plant IDPs might circumvent these limitations in the future and enable single-molecule FRET for in vivo biophysical property determination.

Targeted creation of IDP chimeras in vivo with similar functionality and biophysical signatures might reveal regions and distinct features that are crucial for protein function and open novel avenues for protein design. Chimeric protein approaches were successfully used for folded proteins involved in the development of multiple diseases such as breast cancer, neuroinflammation, Alzheimer’s disease, or addiction [235,236,237]. Chimeras of plant and metazoan IDPs may thus reveal potential targets to improve plant growth under stress conditions or to improve our understanding of key players in human diseases.

## 11. Conclusions

Since protein disorder emerged as a systematically studied field some 20 years ago, the challenge of devising a scheme of classification and functional annotation for the disordered proteome has been widely discussed within the scientific community. Indeed, making meaningful comparisons within the enigmatic realm of disordered proteins that operates outside the classic sequence–structure–function relationship requires navigating between different frameworks of similarity. With this contribution, we want to illustrate the need for curated knowledge transfer across phyla that works in concert with traditional computer-based annotation methods. Describing the functional versatility of IDRs requires a broad and integrative approach that must include evolutionary and structural, as well as, functional and biophysical considerations. Beyond this, deciphering the properties that are common or species specific across evolutionary distant organisms can improve our understanding of how IDP function can evolve in diverse biological contexts and how the interplay between protein structure and disorder creates the diverse functional repertoire found in the proteome.

## Figures and Tables

**Figure 1 ijms-21-02105-f001:**
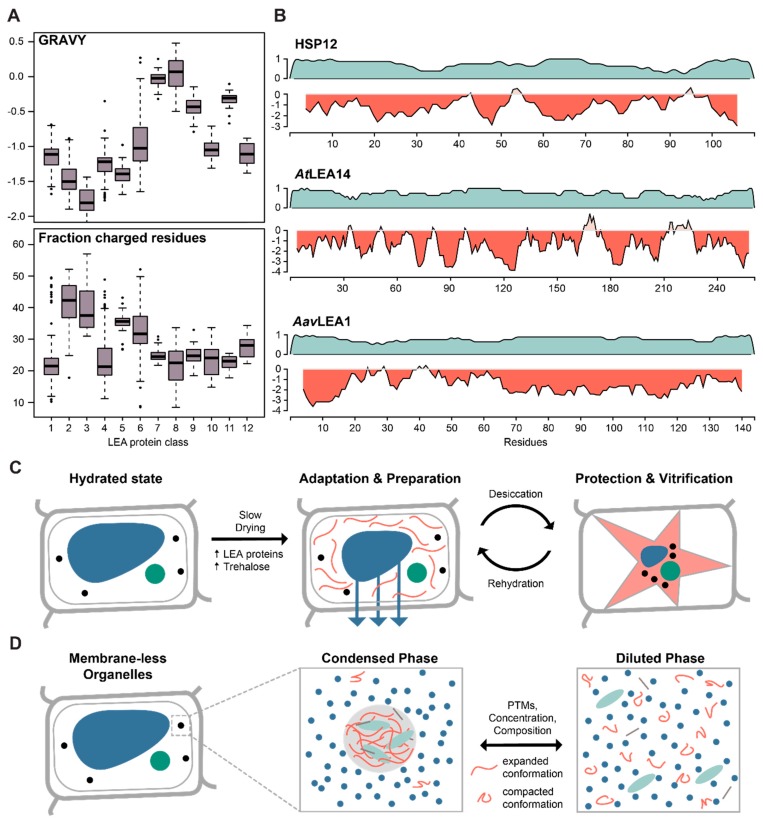
Late Embryogenesis Abundant (LEA) proteins and the role of intrinsically disordered proteins (IDPs) in desiccation and liquid–liquid phase separation. (**A**) Boxplot of the Grand Average of Hydropathy (GRAVY) and the fraction of charged residues (DEKR) calculated for the 12 LEA protein classes, based on statistical analysis of physico-chemical properties and amino acid usage [30]. Adapted and modified with permission from http://forge.info.univ-angers.fr/~gh/Leadb/index.php. (**B**) Disorder prediction (teal) and hydrophobicity plot (red) for hydrophilin representatives from yeast (HSP12), plant (*At*LEA14) and nematode (*Aav*LEA1), all involved in desiccation tolerance. (**C**) Cartoon representation of the molecular processes involved in desiccation. Upon slow drying, trehalose and LEA proteins accumulate in the cell and vitrify, forming an amorphous matrix to stabilize and protect other proteins and membranes during desiccation. (**D**) Cartoon representation of the molecular processes involved in phase separation. Upon reaching a critical concentration, certain IDPs undergo liquid–liquid phase separation to form membrane-less organelles. They undergo a transition from a compacted state in solution to an expanded state in the phase separated droplet. The process can furthermore be modified by other features such as post-translational modifications (PTMs) or protein composition.

**Figure 2 ijms-21-02105-f002:**
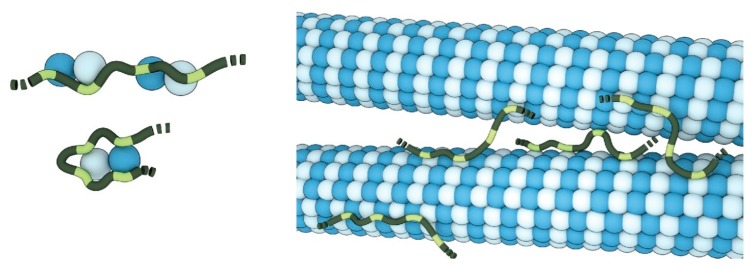
Disordered microtubule-binding proteins. The dynamic nature of the CC1- and Tau-binding (dark- and light-green) behaviour suggests that both might be able to bind multiple distinct tubulin dimers (light- and dark-blue) via their individual binding motifs (light-green), thereby increasing the local tubulin concentration, connecting and stabilizing protofilaments or bundling microtubules. Image by Barth van Rossum.

**Figure 3 ijms-21-02105-f003:**
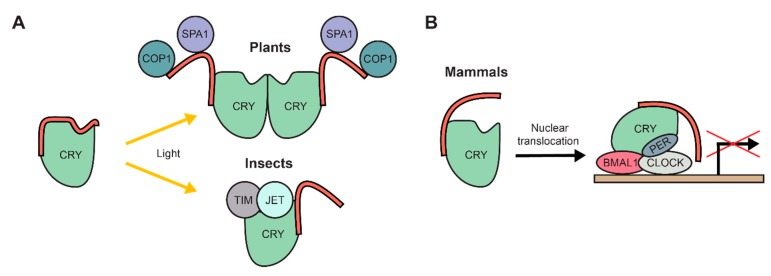
The intrinsically disordered C-terminal extension of cryptochromes (CRYs) has diverse functions. (**A**) Upon light perception through a photolyase homology region (PHR), the C-terminal extension (red) is released from the PHR and acquires an exposed conformation. In plants, the C-terminal extension directly binds to partner proteins, e.g. COP1 and SPA1. In insects it is not involved in binding to proteins, instead the PHR binds to partner proteins, e.g. TIM and JET. (**B**) In mammals, the C-terminal extension is responsible for translocation of CRYs to the nucleus, where they are involved in regulating gene expression as part of the central circadian clock regulation complex together with proteins like BMAL1, CLOCK and PER.

**Figure 4 ijms-21-02105-f004:**
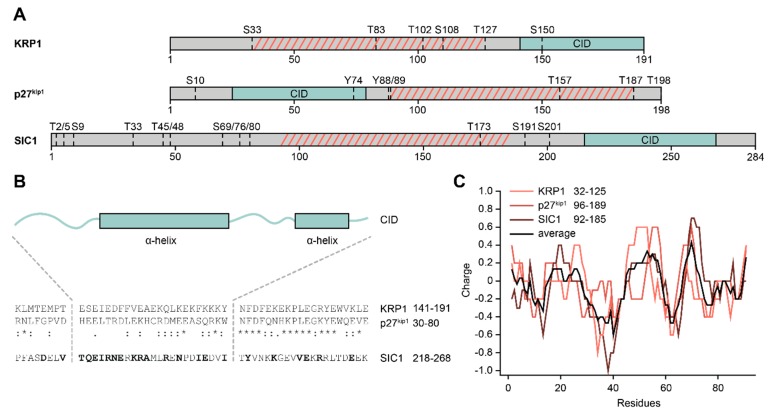
Features of cyclin-dependent kinase inhibitors (CKIs). (**A**) Overview of general CKI architecture across organisms. *Arabidopsis* KRP1, human p27^kip1^, and yeast SIC1 all share a large intrinsically disordered region (grey) that shows no sequence conservation. A sequentially conserved inhibitory domain (CID; green) is located at different positions in the proteins. Dotted lines denote known (p27^kip1^, SIC1) or predicted (KRP1; NetPhos 3.1 score > 0.9) phosphorylation sites in the IDR. (**B)** Sequential comparison of the conserved CID. KRP1 and p27^kip1^ are sequentially conserved, while SIC1 only shows similar amino acids (bold letters) in several positions to either protein. Two alpha helices (boxes) comprise the binding domain of the CIDs, as shown by the crystal structure of p27^kip1^, modelling of SIC1, or prediction with JPRED for KRP1. (**C**) All three CKIs show conserved, charge-based features in the IDR region between important phosphorylation sites (highlighted by hatched, red lines in (**A**).

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
