# Peer review of "Common Functions of Disordered Proteins across Evolutionary Distant Organisms"

_ijms, 2020, doi:10.3390/ijms21062105_

Round 1

Reviewer 1 Report

The review is clear and well written. It gives an overview of the functions of IDPs in plants showing different examples and discuss the similarities with other kingdoms. But authors should revise and complete some details before publication.

  1.  The title should be revised because 'disorder' as a such is not a clear term to refer a protein region with a specific function. The 'disorder' per se has not a function although in most papers on this topic is often written. I consider that this is a conceptual mistake.
  2.  p. 2. The distinction among conserved/identical, similar, non-conserved/variable disordered/ductile regions should be clarified and discussed. For that refers to Bellay et al. Genome Biology (2011); Yruela et al. Front. Plant Sci. (2018)
  3.  p. 2  Authors should also discuss recent papers concerning the function of LEA proteins by Cuevas-Velazquez et al. Plant Signal Behaviour (2017); Artur et al. Front. Plant Sci. (2018); Koubaa et al. Scientific reports (2019). Also evolutionary aspects by Mertens et al. Applied and Environmental Microbiology (2018).
  4.  p. 8 and 10 Discuss the relevance of the increase of IDRs in transcription factors involved in cell cycle during evolution in different kingdoms, see Yruela et al. Genome Biol. Evol. (2017)

Author Response

We thank the reviewers for the thorough review and answered their questions below. Furthermore, we additionally corrected minor phrasing issues and typos.

Reviewer 1)

The review is clear and well written. It gives an overview of the functions of IDPs in plants showing different examples and discuss the similarities with other kingdoms. But authors should revise and complete some details before publication.

The title should be revised because 'disorder' as a such is not a clear term to refer a protein region with a specific function. The 'disorder' per se has not a function although in most papers on this topic is often written. I consider that this is a conceptual mistake.

We changed the title to “Common Functions of Disordered Proteins across Evolutionary Distant Organisms” to make this point more clear.

We furthermore changed the headers of the individual paragraphs to fit the title and added three important publications in the liquid phase separation paragraph:

“Moreover, features like prion-like domains are important predictors of LLPS in RNA-binding proteins (Vernon and Forman-Kay 2019). In Arabidopsis more than 500 proteins were predicted to carry such domains (Chakrabortee et al. 2016). Indeed, the plant-specific Flowering Locus CA and FLL2 proteins have recently been shown to be in vivo regulators of LLPS within the autonomous flowering pathway of Arabidopsis (Fang et al. 2019).”

p2. The distinction among conserved/identical, similar, non-conserved/variable disordered/ductile regions should be clarified and discussed. For that refers to Bellay et al. Genome Biology (2011); Yruela et al. Front. Plant Sci. (2018) p. 2 Authors should also discuss recent papers concerning the function of LEA proteins by Cuevas-Velazquez et al. Plant Signal Behaviour (2017); Artur et al. Front. Plant Sci. (2018); Koubaa et al. Scientific reports (2019). Also evolutionary aspects by Mertens et al. Applied and Environmental Microbiology (2018).

We discussed and cited all mentioned publications at appropriate places (highlighted in the text). Instead of citing Cuevas-Velazquez et al. Plant Signal Behaviour (2017), we cited the original paper from 2016 (not the 2017 addendum version). Furthermore, we added the following sentences (Bellay et al. 2011 was already cited in the manuscript (citation 12 in the submitted version):

“Overall, genome duplication events seem to influence the distribution of IDRs within genomes, as the amount of identical paralogous IDRs positively correlates with the number of chromosomes (Yruela et al. 2018).”

“Some bacterial and plant LEA proteins also carry folded domains like the Water stress and Hypersensitive response (WHy) domain, which is involved in the desiccation response and may originate from an ancestral domain in Archaea (Ciccarelli and Bork 2005; Mertens et al. 2018).”

p8 and 10 Discuss the relevance of the increase of IDRs in transcription factors involved in cell cycle during evolution in different kingdoms, see Yruela et al. Genome Biol. Evol. (2017)

We discussed the mentioned paper by adding the following sentence:

“The disorder content of TFs may be linked to the capacity of to establish more complex gene regulatory networks in multicellular organisms (Yruela et al. 2017).”

Reviewer 2 Report

The review paper provides in-depth analysis and presentation of the functions of intrinsically disordered proteins (IDPs) in plants. Considering that IDPs lack a well-defined structure and thus fall outside the scope of the structure-function paradigm, it is important and interesting to find out and understand the roles of IDPs in plants. Only a few minor points need to be addressed.

1. Figure 1B; the unit of y-axis of disorderness should be labeled.

2. Figure 2; tubulin dimers and IDPs (such as CC1 and Tau) should be labeled.

3. Figure 3 was never cited in the main text. In Figure 3 legend, …acquires an exposed confirmation… confirmation should be conformation.

4. Under the subtitle “Disorder in cell cycle regulation”, citation of Figure 2A, 2B, and 2C should be 4A, 4B, and 4C.

Author Response

We thank the reviewers for the thorough review and answered their questions below. Furthermore, we additionally corrected minor phrasing issues and typos.

Reviewer 2)

The review paper provides in-depth analysis and presentation of the functions of intrinsically disordered proteins (IDPs) in plants. Considering that IDPs lack a well-defined structure and thus fall outside the scope of the structure-function paradigm, it is important and interesting to find out and understand the roles of IDPs in plants. Only a few minor points need to be addressed.

  1. Figure 1B; the unit of y-axis of disorderness should be labeled.

We added a scale to the axis ranging from 0 to 1.

  1. Figure 2; tubulin dimers and IDPs (such as CC1 and Tau) should be labeled.

We added the colours of the respective components to the figure legend to make this point more clear.

  1. Figure 3 was never cited in the main text. In Figure 3 legend, ...acquires an exposed confirmation... confirmation should be conformation.

We are sorry for this mistake and now cited it in the text.

  1. Under the subtitle "Disorder in cell cycle regulation", citation of Figure 2A, 2B, and 2C should be 4A, 4B, and 4C.

We changed the citations and they are now all in the correct order.